# A New Model Organism to Investigate Extraocular Photoreception: Opsin and Retinal Gene Expression in the Sea Urchin *Paracentrotus lividus*

**DOI:** 10.3390/cells11172636

**Published:** 2022-08-24

**Authors:** Periklis Paganos, Esther Ullrich-Lüter, Filomena Caccavale, Anne Zakrzewski, Danila Voronov, Inés Fournon-Berodia, Maria Cocurullo, Carsten Lüter, Maria Ina Arnone

**Affiliations:** 1Department of Biology and Evolution of Marine Organisms, Stazione Zoologica Anton Dohrn, 80121 Naples, Italy; 2Museum für Naturkunde, Leibniz Institute for Evolution and Biodiversity Science, 10115 Berlin, Germany

**Keywords:** extraocular photoreception, sea urchin, scRNA-seq, *opsin*, *pax6*

## Abstract

Molecular research on the evolution of extraocular photoreception has drawn attention to photosensitive animals lacking proper eye organs. Outside of vertebrates, little is known about this type of sensory system in any other deuterostome. In this study, we investigate such an extraocular photoreceptor cell (PRC) system in developmental stages of the sea urchin *Paracentrotus lividus*. We provide a general overview of the cell type families present at the mature rudiment stage using single-cell transcriptomics, while emphasizing the PRCs complexity. We show that three neuronal and one muscle-like PRC type families express retinal genes prior to metamorphosis. Two of the three neuronal PRC type families express a rhabdomeric opsin as well as an echinoderm-specific opsin (echinopsin), and their genetic wiring includes sea urchin orthologs of key retinal genes such as *hlf*, *pp2ab56e*, *barh*, *otx*, *ac/sc*, *brn3*, *six1/2*, *pax6*, *six3*, *neuroD*, *irxA*, *isl* and *ato*. Using qPCR, in situ hybridization, and immunohistochemical analysis, we found that the expressed retinal gene composition becomes more complex from mature rudiment to juvenile stage. The majority of retinal genes are expressed dominantly in the animals’ podia, and in addition to the genes already expressed in the mature rudiment, the juvenile podia express a ciliary opsin, another echinopsin, and two Go-opsins. The expression of a core of vertebrate retinal gene orthologs indicates that sea urchins have an evolutionarily conserved gene regulatory toolkit that controls photoreceptor specification and function, and that their podia are photosensory organs.

## 1. Introduction

Over the last decade, there has been a surge of interest in the evolution and mechanisms of extraocular photoreception. Little is known about those mechanisms and their players in other deuterostome phyla outside of vertebrates. Sea urchins, marine deuterostomes, have drawn attention due to their variety of photosensitive behaviors despite their apparent lack of distinct eye organs and a centralized brain.

Molecular fingerprinting of certain cell types allows us to investigate the evolution and development of extraocular photoreceptors from a molecular perspective. It has been shown that across many metazoan phyla, photosensitive proteins which drive the phototransduction cascade inside the primary receptors (opsins) and retinal gene orthologs are conserved (for review see [1]). While some genes, like *pax6*, *otx* and genes of the *six* family are almost universally expressed in precursors of PRCs across the animal kingdom [1,2,3] others are expressed to determine a specific PRC type fate [4,5]. Those include *atonal/math5*, *brn3/pou4f* and *barh* for rhabdomeric opsin expressing PRCs, such as retinal ganglion cells in vertebrates, or rhabdomeric opsin (r-opsin) expressing PRCs in the eyes of the polychaete *Platynereis* [6]. In contrast, *rx* is a retinal transcription factor restricted to progenitors of ciliary opsins expressing PRCs such as rods and cones in the developing vertebrate retina [7,8].

The genome of *Strongylocentrotus purpuratus* was found to contain an unexpectedly large repertoire of opsin and retinal gene orthologs [9,10]. Due to the fact that sea urchin podia have been shown to be photosensitive in many species [11,12] and have a large number of neurons and primary sensory receptors with unknown functions [13], they have been hypothesized to be photosensory organs [14,15,16]. Burke and colleagues [9] discovered three opsins while performing transcription analysis on *S. purpuratus* podia, the ciliary opsin Sp-opsins1, the r-opsin Sp-opsin4 and an opsin of unclear identity Sp-opsin5. The molecular cascade driving the specification and differentiation of PRCs depends on the transcription factors active during PRC development. Evidence for the expression of key retinal transcription factors was reported in the same study by Burke et al., in which the authors found expression of the retinal genes *pax6*, *atonal*, *six3 isl1*, *rx*, *brn3*, *neuroD*, *blimp1*, and *barh*. On the other hand, Raible et al. [10] found *Sp-opsin1*, *Sp-opsin2* (which they suggested was an echinoderm novel opsin), *Sp-opsin3.1* (a Go-Opsin), and *Sp-opsin4* transcripts at high levels in podia.

In search of sea urchin “eyes”, spatial analysis of sea urchin opsins and retinal orthologs in the following focused on expression patterns of rhabdomeric and ciliary opsins, the two major opsins expressed in visual photoreceptors across the animal kingdom [1]. In *S. purpuratus*, Ullrich-Lüter et al. [16] found expression of *Sp-opsin4* in the animals’ podia, which are locomotory and sensory appendages that the sea urchin can extend and withdraw, and in particular in specific cell clusters embedded in small skeletal cavities at the base of those appendages. Ullrich-Lüter et al. [17] later discovered that the second candidate opsin type potentially involved in vision, Sp-Opsin1 is expressed throughout the animals’ epidermis, including podia, pedicellariae, and spines. Agca et al. [14] reported three opsin genes in *S. purpuratus* adult podia via qRT-PCR analysis, namely *opsin4*, *opsin1* and *opsin5*. They also showed that expression levels of all three opsin types are generally higher in the disc than in the stalk region of the podia and that *pax6* is expressed inside the rim of the podial disc. Lesser et al. [15] reported a “rhabdomeric-like” opsin in the podia of the green sea urchin *Strongylocentrotus droebachiensis*, although their phylogenetic analysis showed that this opsin in fact clusters between rhabdomeric and ciliary opsins and varies very little from the *S. purpuratus Sp-opsin5*. *Sp-opsin5* was already suggested to be a novel sea urchin-specific opsin by Raible et al. [10], a finding later confirmed by D’Aniello et al. [18] who accordingly termed this echinoderm-specific opsin “echinopsin”. 

D’ Aniello et al. [18], screening multiple genomic and transcriptomic data sets from all five extant echinoderm subgroups, discovered 97 opsin sequences in total, 33 of which were found in sea urchins. According to their analysis, sea urchins express at least seven different opsin genes: ciliary opsin (*C-opsin* or *opsin1* [19]), two Go-opsins (*opsin3.1* and *3.2* [20,21]), rhabdomeric opsin (*r-opsin* or *opsin4* [19]), echinopsin A (*opsin2*), echinopsin B (*opsin5* [18]), peropsin (*opsin6* [19]) and RGR-opsin (*opsin7* [19]).

Aside from the two potentially visual opsins, the expression patterns and potential functions of other opsin types and retinal gene orthologs in post metamorphic sea urchins remains largely unknown. In adult sea urchins, *pax6* is found throughout the entire podia and the region of their origin [14,16,22]. Ullrich-Lüter et al. [16] discovered *pax6* expression in areas adjacent to Opsin4 positive cells in the podial disc, but not overlapping Opsin4 expression in adult *S. purpuratus* podia. Lesser et al. [15] immunostained histological sections and reported the presence of Pax6 protein in pigment cells within the podial disc’s perforated ossicles. Acga et al. [14] discovered that seven retinal transcription factors including *pax6*, *brn3*, *six3*, *neuroD*, *rax*, *isl1* and *ato6* are found in the disc and stalk regions of the podia of *S. purpuratus*. Mao et al. [23] showed *brn3* expression accompanying skeletal elements in the podial disc region. They also found *pax6* highly expressed inside the nerves of the podial stalk and a lower level of expression on the outer rim of the podial disc, where many receptor cells reside. *pax6* and *brn3* expression occurs in close proximity but does not seem to overlap. Byrne et al. [24] examining genes involved in the *pax*, *six*,, *eya*, *dach* network in young juveniles of the direct developing sea urchin *Heliocidaris erythrogramma* found *six1/2* and *six3* expression in the circumoral nerve ring and *six3* expression in the longitudinal podial nerve. In addition, they discovered a dynamic *pax6* expression pattern in developing juvenile podia.

Although in situ hybridization has proven to be a useful tool for studying the expression patterns of genes, this type of analysis, which provides information about the spatial and temporal expression of gene and transcription factor products, has limitations. Only a limited number of genes can be examined concurrently at any given time. In contrast, we can disentangle the molecular signature of a specific cell type with much greater granularity using single cell RNA sequencing (scRNA-seq). This is critical to our understanding of the function and evolutionary origins of retinal gene orthologs and opsins in the context of our research. Traditionally, data on the molecular composition of cell types was obtained through gene candidate surveys, which resulted in the identification of differentially expressed mRNAs and proteins in different cell types. ScRNA-seq now allows for the generation of thousands of single cell transcriptomes that reflect both the bulk and fine differences between different cell types and cell states. This droplet-based technique’s workflow includes dissociating a whole organism, organ, or tissue into single cells, which are then encapsulated in droplets, cDNA library construction, and sequencing. The key feature of this method is that the mRNAs of each encapsulated cell are uniquely labeled, allowing for the correlation of gene expression and distinct cell populations. Cells with similar expression profiles are grouped into clusters that correspond to distinct cell type families, cell types, or cell states using computational analysis. ScRNA-seq has been used successfully to describe and identify novel cell types in a wide range of organisms, including sponges [25], cnidarians [26,27], and echinoderms [28,29,30], as well as mammals [31]. In addition, scRNA-seq has been used to decipher complex evolutionary relationships between cell types. Single cell transcriptomics, for example, has been used to describe evolutionary conserved and species-specific gene expression patterns as well as differentiation mechanisms during human and mouse retinal development [32]. 

In this study we chose *P. lividus* to study extraocular photoreception because its genome is available [33] and it has a relatively short biphasic life cycle, approximately 5 weeks from fertilization to metamorphosis [34,35], allowing us to investigate different developmental stages in rapid succession. Furthermore, behavioral testing in *P. lividus* according to the procedures used by Kirwan et al. [36] in the sea urchin, *Diadema africanum*, has recently demonstrated that these sea urchins possess spatial vision sensu stricto, too (data not shown). Although sea urchins are invertebrates, the tissues that may be involved in photoreception have a complex composition, with a wide range of cell types and neurons. Here we used scRNA-seq to gain an overview of all active genes at a given stage of development. Subsequent in situ analysis of protein and mRNA expression of retinal genes allowed for the precise expression allocation, as well as timing of transcription factors and terminal differentiation genes up- and downregulation. Overall, we provide a thorough characterization of the *P.lividus* PRCs as well as evidence for an increasing PRC complexity during *P. lividus* development. Finally, we show that *P. lividus* PRCs utilize an evolutionarily conserved molecular genetic toolkit, thus making it an interesting animal model for learning about the evolution of extraocular photoreceptors in a non-chordate deuterostome.

## 2. Materials & Methods

### 2.1. Animal Husbandry and Culture of Embryos, Larvae, and Juveniles

Adult *P. lividus* individuals were collected from the Gulf of Naples and maintained in circulating seawater aquaria at Stazione Zoologica Anton Dohrn (Naples, Italy). Gametes were obtained by vigorous shaking of the animals. Embryos and larvae were reared in incubators at 18 °C in Mediterranean filtered seawater (FSW) and using a 12/12 h light/dark cycle. The salinity of the FSW was 37 ppt. Larvae were kept at a concentration not higher than 5 larvae/mL in 2 L beakers. Larval cultures were maintained by exchanging half of the water with fresh FSW 2 times per week. After the 2 days post fertilization (dpf) pluteus stage, larvae were fed 3 times per week with the unicellular algae *Dunaliella* sp. at a concentration of 1000 cells/mL. Once the animals reached the juvenile stage, they were kept separately from the competent larvae and were fed with *Ulva lactuca*. 

### 2.2. Rudiment Microdissection and Dissociation 

Mature rudiments originating from two biological replicates were used. Per biological replicate, three rudiments were manually collected through dissection by using tungsten needles under a stereoscope. Each rudiment was dissected out of the larva, while larval tissues were discarded. For dissociation of the rudiments into single cells, seawater was removed, and dissected rudiments were resuspended in Ca^2+^ Mg^2+^-free artificial sea water containing 1% Protease XIV (Sigma-Aldrich, Burlington, MA, USA) 1% and 0.05% Liberase (Sigma-Aldrich, MA, USA). Dissociation was performed at 37 °C for 40 min, mixing gently via pipette aspiration every 2 min. All the debris were removed by passing the suspension through a 0.22 µm mesh, the dissociated cells were spun down at 700 g for 5 min and washed several times with Ca^2+^ Mg^2+^-free artificial sea water. Cell viability was assessed using Propidium Iodide (ThermoFisher Scientific, Waltham, MA, USA) (1 µg/mL) and Calcein AM (ThermoFisher Scientific, MA, USA) (2 µg/mL) and only specimens with high cell viability were further processed. Single cells were counted using a hemocytometer and diluted according to the manufacturer’s guidelines (10x Genomics).

### 2.3. Single Cell RNA Sequencing and Computational Analysis

Single cell RNA sequencing analysis has been carried out as described in [30]. Briefly, scRNA-seq was performed using the 10x Genomics Chromium technology. Specimens from two independent biological replicates were loaded on the 10x Genomics Chromium Controller. Single-cell cDNA libraries were prepared using the Chromium Single Cell 3′ Reagent Kit v3.1 (10x Genomics, Pleasanton, CA, USA) and the resulting libraries were sequenced by GeneCore (EMBL, Heidelberg, Germany) for 75 bp paired end reads (Illumina NextSeq 500, CA, USA). The genomic index was made in Cell Ranger using the *P. lividus* genome [31]. Cell Ranger (10x Genomics, CA, USA) output matrices were used for further analysis using the Seurat v4.0.5 R package [37]. Genes that are transcribed in less than three cells and cells that have less than a minimum of 200 transcribed genes were excluded from the analysis. Datasets were normalized and variable genes were found using the Variance stabilizing transfer (VST) method with a maximum of 2000 variable features. Data integration was performed via identification of anchors between the two different objects. Datasets were scaled and principal component (PCA) analysis was performed. A Sharing Nearest Neighbor (SNN) graph was computed with 32 dimensions (resolution 1.0). Uniform Manifold Approximate and Projection (UMAP) was used to perform clustering dimensionality reduction. The final object consisted of 2036 cells. Cluster markers were found using the genes that are detected in at least 0.01 fraction of minimum percent (min.pct) cells in the two clusters. Sub-clustering analysis was performed by selecting the photoreceptor 2 and photoreceptor 3 cell type families and performing similar analysis as described above. For photoreceptor 2 and photoreceptor 3 generated sub-clustered objects, SNN graphs were computed with 8 and 6 dimensions, respectively (resolution 1.0).

### 2.4. Whole Mount Fluorescent In Situ Hybridization (FISH)

FISH was carried out as described in [38]. Antisense probes were synthesized as described in [39]. Probes were transcribed from linearized DNA and labeled during transcription using Digoxigenin or Fluorescein RNA labeling mix solutions (Sigma-Aldrich, MA, USA), following the manufacturer’s instructions. Probes for *Pl-pax6* and *Pl-six3* were produced as previously published [40]. Primer sequences used for cDNA isolation and probes synthesis are in Appendix A. Specimens were imaged using a Zeiss LSM 700 confocal microscope.

### 2.5. Immunohistochemistry (IHC)

IHC was carried out as described in [39]. Briefly, specimens were fixed in 4% paraformaldehyde (PFA) in filtered sea water (FSW) for 15 min at room temperature (RT). FSW was exchanged with 100% methanol for 1 min at RT, washed multiple times with phosphate buffer saline with 0.1% Tween 20 (PBST) and incubated blocking solution containing 1 mg/mL BSA (Sigma-Aldrich, MA, USA) and 4% sheep serum (Sigma-Aldrich, MA, USA) in PBST for 1 h. Primary antibodies were added in the appropriate dilution and incubated overnight (ON) at 4 °C. 1E11 recognizing Synaptotagmin b (Syt1), a gift from Dr. Robert Burke [41], was used to label the nervous system (1:20), acetylated tubulin (Sigma-Aldrich, MA, USA) to mark ciliated structures and the nervous system (1:200), Sp-Opsin1 (1:50) and Sp-Opsin4 (1:50) were used to label PRCs [16] and anti-Sp-MHC, myosin heavy chain, (1:100-PRIMM, Italy) was used to label muscle cells. Specimens were washed with PBST (5 times) and incubated for 1 h with the appropriate AlexaFluor (ThermoFisher Scientific, MA, USA) secondary antibody diluted 1:1000 in PBST. Samples were washed several times with PBST and imaged using a Zeiss LSM 700 confocal microscope.

### 2.6. Quantitative Real Time PCR (qRT-PCR)

The relative expression profiles of the *P. lividus* opsin genes at juvenile stage were defined by qRT-PCR analysis. Primer sequences used for the qPCR are in Appendix A. Total RNA was extracted from three biological replicate samples, each of them made by a pool of 5 specimens, reared as described in Section 2.1. and using the RNeasy Plus Mini Kit (Qiagen, Germany). RNA extraction was performed by trituration using a T25 ULTRA-TURRAX. Using the SuperScript VILO cDNA Synthesis Kit (ThermoFisher Scientific, MA, USA), 100ng of RNA was retro-transcribed into cDNA and was used undiluted. Composition of the PCR reactions, cycling conditions and the analyses of the results were performed as described in [42]. The 2^−∆∆Ct^ method was used to calculate the relative gene expression. Ribosomal protein L17 (Rpl17), expressed at a constant level during development, was used as a reference gene for the normalization of each gene expression level [43]. Since *Pl-opsin 3.1* appeared to have the lowest expression levels, for aesthetic reasons the expression of the rest of opsin genes were plotted in comparison to *Pl-opsin3.1*. For statistical analysis, we used the GraphPad Prism software employing the paired t-test. Statistical significance cut-off criteria were set at *p* < 0.05.

## 3. Results

### 3.1. Generating a Cell Type Atlas of the Mature Rudiment Stage with Single-Cell Transcriptomics

To gain insight into the photoreceptor repertoire and the molecular fingerprint prior to metamorphosis, we performed scRNA-seq on *P. lividus* mature rudiments. Larvae originating from two independent biological replicates were cultured and collected at the eight-arm competent larval stage; the rudiment was manually dissected out of the larva, dissociated into single cells and single cell libraires were generated. Our computational analysis revealed 20 transcriptomically distinct cell clusters, corresponding to an individual cell type or closely linked cell type families (Figure 1).

We used the large number of cell type gene markers for sea urchins that have been shown to successfully label distinct cell types throughout sea urchin development to identify several putative cell type families [29,30,44]. In detail, using known musculature gene markers (*trop1*, *mlckb*, *foxC*, *myoD*), skeletal (*alx1*, *sm37*, *sm50*), pigment cells (*pks1*) and immune system (*macpfA2*) markers, we were able to identify at least three muscular, one skeletal and one immune system-related populations (globular cell-like), respectively. In the case of *pks1* positive cell types, expression of *pks1* was found in a large cohort of cell type families, suggesting that pigmentation can be a feature shared by more than one cell type family. Interestingly, plotting gene markers labeling distinct domains of the digestive tract of the larva resulted in the recognition of three cell type families corresponding to the posterior gut (*pdx1*, *cdx1*, *endo16* positive), the stomach (*manrC1a* positive) and to a specialized exocrine cell type populating the upper part of the stomach (*cpa2 L*, *serp2–3*, *amy2* positive). The presence of those cell populations in the single cell datasets, despite the microdissection of the rudiment out of the larva, is in line with previous studies demonstrating that these regions are reused to form the digestive system of the juvenile [45]. Furthermore, using the neuronal markers synaptotagmin 1 (*syt1*), choline O-acetyltransferase (*chaT*), histidine decarboxylase (*hdc*) and tyrosine hydroxylase (*th*), we were able to identify approximately five neuronal cell type families, each with a distinct molecular fingerprint. Finally, plotting the average expression of *fcolI/II/III* and *6afcol*, both involved in collagen production and known to label the blastocoelar cells of the sea urchin larva, allowed us to recognize cell type families that are able to produce this biomolecule also in the juvenile stage. Based on the presence of all the aforementioned cell types, we gained confidence that our datasets are able to reconstruct a great amount of the cell type diversity at this developmental stage.

### 3.2. Investigating the Molecular Fingerprint of Prcs in Mature Rudiment

Then, we set out to accomplish our main goal, which was to investigate the photoreceptor composition of the sea urchin *P. lividus*. To this end we took advantage of the opsin phylogeny performed by D’Aniello and colleagues [18] and classified all opsins found accordingly (Appendix A). When all the opsin genes encoded in the sea urchin genome were plotted, only two opsin-type PRCs and in total four distinct cell type families were found in the mature rudiment stage (Figure 1C). One of these families corresponds to a cell population expressing Opsin2 and histidine decarboxylase (Photoreceptor cells 1), two of these families (Photoreceptor cells 2 and 3) correspond to neurons (*syt1* positive) co-expressing *opsin2* and *opsin4*, while the last one (Muscles 3) resembles a muscle-like cell cluster expressing exclusively *opsin2*.

To address how similar/different the two PRCs double *opsin2/opsin4* populations were, we investigated the expression profiles of the differentially expressed genes that are dictating the clustering analysis: the marker genes. Using a heatmap depicting the marker genes of both *opsin2/opsin4* positive cell type families (Figure 2A) it is evident that the two populations have a very distinct molecular signature. This is also evident from the Venn diagram of these marker genes, which indeed shows only 37 shared markers between the two cell type families. Among these 37 genes, apart from *Pl-opsin4*, there are the Notch signaling component *Pl-notch2*, the transcription factor *tbx3*, typical photoreceptor markers of photoreceptors and the neuronal markers myoactive neuropeptide NGFFFamide (*Ngfffap*) and Synaptotagmin *syt4*. Notably, these 37 genes also represent a unique molecular signature of these PRCs in the atlas, as shown in the dotplot reported in Figure 2B, thus suggesting that the shared molecular signature stands alone in respect to the rest of the cell type families.

To further characterize the photoreceptor cell type families and to ensure that the co-expression of *opsin2* and *opsin4* is indicative of a distinct cell type and not closely related cell types, we performed sub-clustering analysis of the two *opsin2*/*opsin4* double positive cell type families (Figure 3). As also previously shown in [30], sub-clustering analysis is sufficient to identify unique cell types and cell states from an apparently uniformal cluster or cell type family. In doing so, we discovered that the two *opsin2*/*opsin4* double positive cell type families can be subdivided to three and two distinct clusters, respectively. In total, two sub-clusters appear to be *opsin2*/*opsin4* double positive, while only one seems to express solely *opsin4*. To investigate the molecular signature of the three different PRC putative cell types, we plotted known orthologs of vertebrate retinal genes (Appendix A) encoded in the sea urchin genome [9,10,46]. Overall, our data show a broad expression of these genes in the various PRCs sub-clusters (Figure 3B). Surprisingly, the two *opsin2*/*opsin4* double positive sub-clusters contain transcripts encoded by different combinations of retinal genes suggesting either the presence of two diversified *opsin2*/*opsin4* double positive cell types or different states of their development. In detail, the *opsin2*/*opsin4* double positive sub-cluster originating from the photoreceptor 2 cluster of our analysis contains transcripts for *hlf*, *glass*, *otx*, *brn3*, *dach*, *pax6*, *neuroD*, *glis and isl*, while the *opsin2*/*opsin4* double positive sub-cluster originating from the photoreceptor 3 original cluster contains cells expressing *pp2ab56e*, *ac/sc*, *brn3*, *atbf1*, *irxA* and *ato.* On the other hand, one cluster, coming from the sub-clustering analysis of the photoreceptor 2 original cluster, expresses only *opsin4* in combination with the retinal gene orthologues of *pp2ab56e*, *barh*, *otx*, *ac/sc*, *six1/2*, *atbf1*, *irxA* and *ato.* While with the current data, the question whether differences in gene expression are indicative of distinct cell types or developmental cell states cannot be addressed, it is striking that all sub-clusters have a molecular signature resembling a proper vertebrate photoreceptor cell.

### 3.3. Spatial and Temporal Expression of Opsin4, Pax6, Neurod and Six3 during Juvenile Development

#### 3.3.1. Opsin4

We found Opsin4 protein and mRNA in cells along the discs of primary podia as early as the mature rudiment stage (Figure 4A,B). Both expression of mRNA and protein presence persist through metamorphosis into the juvenile. We also found a second cluster of rhabdomeric opsin positive PRCs at the base of each tube foot in specimens that had already grown the first pair of bilaterally symmetrical podia after the initial singular primary podium (Figure 4C). These cells are arranged into bilaterally symmetrical clusters that are anatomically located between the podia and the neighboring primary spines. The specificity of the rhabdomeric opsin antibody in the sea urchin *P. lividus* was demonstrated by protein and mRNA co-localization (Figure 4D–F). mRNA is restricted to the cell body and thus absent in the dendritic part of the cells inside the basal and disc podial PRC clusters. While there is a clear overlap between protein and mRNA expression in the apical portion of the cell body in the disc PRCs, this distinction is less pronounced in the basal PRC cluster. Opsin protein but no mRNA is present in the apical cell regions, with their shape resembling microvillar extensions containing the photoreceptive protein (Figure 4F). Each cluster of PRCs at the podial bases consists of one to four cells at the juvenile stage shown (Figure 4F). Each juvenile podial disc contains between one and seven PRCs (Figure 4D,G).

The application of an anti-synaptotagmin (1e11) antibody [41] in further developed juveniles of *P. lividus* reveals the innervation of podia and spines, as well as the animals’ internal nervous system—the five radial nerves interconnected via the circumoral nerve ring (Figure 4G,H). Antibody staining for MHC shows the prominent muscles of the spine bases, as well as the muscle layer inside the podial lumen. Each of the two clusters of Opsin4 + PRCs at the base of each podium now features about 12 cells (Figure 4I). Before passing through the pores of the calcite sea urchin skeleton, neural projections of the primary spines connect with those of the podia (Figure 4J). The Opsin4 + PRC clusters are associated with these neural projections (Figure 4J–L). 

Podial disc PRCs are present in all primary podia at this stage of development, but only a few can be found in the smaller secondary podia that are still developing (Figure 4G). In addition, comparing the overall morphology of base and disc PRCs, we found similarities in their cellular regionalization. In both cases their cell bodies contact projections of the nervous system, while their apical cell portion featuring membrane enlargements faces the opposite side (Figure 4F,I). This distinct polarity suggests that disc and base PRCs are projecting axons into the network of neuronal projections that run into the podial nerve and through the podial pores, respectively. However, with the antibodies available at the time, we were unable to characterize those PRC axons.

#### 3.3.2. Pax6, NeuroD and Six3

The entire field, from which the primary podia were to emerge, was found to be *pax6* positive in the mature rudiment (Figure 5A). The expression of *pax6* was most prominent in the terminal disc of the primary podia of larvae competent for metamorphosis (Figure 5B), with Opsin4 protein already being present at the bases of the primary podia at this developmental stage. In the primary podia of the juvenile, almost no *pax6* expression was found. Instead, the secondary podia displayed high levels of *pax6* expression on their oral side, where the podial ganglion resides. The mRNA was more uniformly expressed throughout the secondary podia, and most dominant in the stalk region, later in development (Figure 5D,E). Tissues expressing *pax6* are close to Opsin4-positive PRCs in the oral disc area and at the base of secondary podia at that stage. However, at the cellular level, we found no clear evidence of *pax6* and Opsin4 co-localization in juveniles. 

The transcription factor *neuroD* was first found during the juvenile stage. It is strongly expressed inside the proximal part of the secondary podia discs (Figure 5F). Expression of *neuroD* was also found within the buccal podia. 

We found *six3* to be expressed as early as in the mature rudiment stage (Figure 5G) in the podial discs, as well as the buccal podia of juveniles, which have already grown the first pair of secondary podia. Like *pax6*, at that developmental stage, *six3* has the highest expression in the oral region of the podial discs, where the podial ganglion is located (Figure 5H). Moreover, *six3* is not expressed inside the primary podia at this stage of development, which is consistent with *pax6* expression (Figure 5H,I). The co-localization of *six3* and Opsin4 protein demonstrates that the opsin remains to be expressed within the primary podia, at this stage, whereas *six3* is not (Figure 5I).

### 3.4. Expression Profiles of Opsin Genes in Post-Metamorphic Juveniles

#### 3.4.1. Relative Gene Expression of Opsin1, Opsin2, Opsin3.1, Opsin3.2, Opsin4 and Opsin5

ScRNA-seq applied on mature rudiments was able to successfully identify PRCs and their molecular signature, while at the same time suggesting that only *opsin2* and *opsin4* are expressed at this stage. To further understand the PRCs composition at the juvenile stage, and to overcome technical difficulties related to the dissociation of whole juveniles into single cells, we performed relative gene expression analysis (qPCR) in specimens grown under the same conditions as the specimens used for scRNA-seq. This analysis revealed a more diverse opsin repertoire at this stage of development (Figure 6A). Except for one peropsin (*opsin6*), which could not be investigated due to a technical annotation, the juvenile expresses all opsin types found in *P. lividus* [18]. The relative expression of the various opsin types varies greatly. The mRNA of *opsin2* was discovered to be more than twice as abundant as the second most abundant *opsin4* (Figure 6A). *Opsin5*, was expressed a magnitude lower than the level of *opsin2*, with even lower levels of expression for the remaining opsins (*opsin1*, *opsin**3.1* and *opsin3**.2*) (Figure 6A).

#### 3.4.2. Ciliary Opsin (*opsin1*)

Cells immunoreactive to an antibody targeting Opsin1 were found to be distributed within the entire epidermis of the animals. Opsin1 was detected in the integument, within primary and secondary podia, in buccal podia as well as in spines (Figure 6B). Co-labeling with anti-acetylated alpha tubulin shows large amounts of neuronal fibers spanning the animals’ epidermis. Opsin1was co-localizing within a fraction of those neural fibers, e.g., in podial discs and in the integument around the mouth opening, (Figure 6B) as well as in the spines and spine bases (Figure 6C). Opsin1 positive fibers run through large portions of the epidermis (Figure 6C,D).

#### 3.4.3. Echinopsin (*opsin2*)

*Opsin2* positive cells are most prominent in the juvenile animals’ primary and secondary podia, as well as their buccal podia (Figure 6D). Numerous *opsin2* positive cells are present at the underside of the podial disc. These cells are in proximity to clusters of Opsin4 positive disc PRCs. More *opsin2* positive cells are found at the base of the developing podia, again closely associated with Opsin4 positive PRCs (Figure 6E,F). However, there was no unambiguous co-localization of *opsin2* mRNA and Opsin4 protein in any given cell.

## 4. Discussion

The goal of our research on the developmental stages, before and after metamorphosis, of the Mediterranean sea urchin species *P. lividus* was to gain a developmental perspective on the fate of the animals’ photoreceptors to learn about the evolution of deuterostome extraocular photoreception. 

While sea urchin embryos have been a valuable model for understanding cell type molecular specification and differentiation via gene regulatory networks, research on later developmental stages has been limited due to the complexity of tissues involved in functions such as photoreception. To overcome this challenge, we used scRNA-seq to obtain a more complete molecular fingerprint of cell types expressing retinal gene orthologs required for photoreception. Then, using immunohistochemistry and in situ hybridization, we were able to determine where and when retinal genes are expressed.

### 4.1. Cell Type Identity

As photoreceptors were expected to be found within cell type families featuring a neuronal signature, we analyzed the scRNA-seq data of mature rudiments using neuronal markers such as synaptotagmin [41], choline O-acetyltransferase [42], histidine decarboxylase [47] and tyrosine hydroxylase [48]. Our analysis showed five neuronal cell type families, each with a distinct molecular fingerprint (Figure 1). Three neuronal cell type families expressing opsins were found, two of them with a neuronal signature and expressing *opsin2* and *opsin4*. One additional cell type family was found to express a single opsin, namely *opsin2*, but to feature a non-neuronal muscle-like composition. A growing body of evidence shows opsin expression in non-neuronal tissues in various animals, ranging from cephalopods to fishes [49,50,51,52], but the roles of most of these non-visual, non-neuronal photoreceptors remain elusive. However, some of those opsins are e.g., active in photorelaxation mechanisms of airway smooth muscle in mice [53], as well as in smooth muscle relaxation of the human uterus [54]. Other research found different opsins modulating hair growth in human hair cells [55]. The presence of opsin expressing cell types in sea urchin non-neuronal tissues therefore represents another interesting case study to assess the role of non-visual photoreception in future research.

The two neuronal *opsin2/opsin4* positive cell type families share common neuronal and retinal-like molecular fingerprints. The neuronal fingerprint includes the transcription factor *arxl*, that in vertebrates is important for forebrain development [56], the synaptotagmin gene *syt4l* [28,30] and the neuropeptide NGFFFamide [30,57,58]. Moreover, the two *opsin2*/*opsin4* positive clusters contain transcripts for the retinal marker gene *Pl-opsin4* [59], the notch signaling component *Pl-notch2* [60,61] and the transcription factor *tbx3* [62] which are involved in vision-related morphogenetic and physiological procedures in vertebrates. However, the two clusters share only 37 marker genes (Figure 2), thus indicating divergent developmental paths of those cell types as well as potentially different functions. Plotting retinal gene orthologs encoded in the sea urchin genome [9,10], our data show a broad expression of these genes in various cell type families, although their combinatorial expression seems to be limited to distinct clusters. A core of retinal genes was found to be expressed in one of the two *opsin2*/*opsin4* double positive cell types including *hlf* [63], *pp2ab56e* [64], *barh* [65], *otx* [66], *ac/sc* [67], *brn3* [68], *six1/2* [69], *pax6* [70], *six3* [69], *neuroD* [71], *irxA* [72], *isl* [73], *atbf* [74], *glis* [75], and *ato* [76], sharing this gene repertoire with vertebrate photoreceptor cells. A similar molecular signature is detected in an opsin negative cluster of unknown identity that could potentially reflect newly specified photoreceptor cells. Moreover, five of these genes were found expressed in the second *opsin2*/*opsin4* positive cell type family, while *hlf and pp2ab56e* are a part of the molecular signature of all three neuronal PRCs. Interestingly, the presence of *opsin4* transcripts in two clusters could potentially reflect the two morphologically distinct cell clusters of PRCs shown by in situ hybridization and immunohistochemistry already at this developmental stage (see Figure 4 and Figure 5). Previous studies carried out on *S. purpuratus* larvae demonstrated the presence of an evolutionary conserved gene regulatory module consisting of the transcription factors *rx*, *otx*, *six3* and *tbx2/3* in Go-opsin positive PRCs [77]. Surprisingly, only *otx* transcripts were found in opsin4 positive clusters, while neither *rx* nor *tbx2/3* transcripts were detected in any of the PRCs analyzed in this study. These data suggest that the sea urchin Go-opsin positive and rhabdomeric photoreceptors follow diversified genetic programs. We cannot, however, rule out the potential that these transcription factors operate during PRC specification or differentiation at different developmental times other than the ones sampled in this work. Therefore, we believe future studies are needed to reach a safe conclusion regarding the spatiotemporal expression of the genes involved in the rhabdomeric PRCs gene regulatory network.

### 4.2. Retinal Gene Expression and Its Putative Function

At the level of the specifying transcription factors, developing animal eyes in a wide range of groups share an at least early involvement of *pax6* (for review see [1]), the expression of which we find in both opsin positive cell type families. *Otx/orthodenticle* and members of the *six* family are also involved in the early specification of PRCs in Bilateria [1]. Interestingly, scRNA-seq revealed *pax6*, *otx*, *six1/2* and *six3* expression inside one or several of the *opsin2*/*opsin4* positive cell populations found in the sub-clustering analysis but not in all of them. This finding could represent different expression profiles of the same cell type caught at different stages of specification/differentiation. 

ScRNA-seq also showed expression of genes specific for rhabdomeric PRCs types. *Atonal*, driving eye precursor cells into rhabdomeric PRC fate in *Drosophila* [78,79] as well as retinal ganglion cell formation in mice [76], was found in *opsin2*/*opsin4* positive cell type clusters, as were its putative downstream transcription factors *brn3* and *barh* (both involved in specification of rhabdomeric ganglion cells in vertebrates [1]). The sea urchin *brn3* ortholog (*pou4f2*) has also been shown to be a functional replacement for *pou2f4*-null mice, where it restores retinal ganglion cells [23]. The expression of these genes involved in the formation, specification, and differentiation of rhabdomeric PRCs supports our finding that the rhabdomeric PRC system is already in place by the time of metamorphosis, as evidenced by the discovery of r-opsin positive PRCs inside primary podia of the mature rudiment. 

Another retinal transcription factor, *neuroD*, critical for the formation of different PRC types in the vertebrate retina [80] was also expressed in one of the *opsin2/opsin4*+ cell type families as the sub-clustering analysis showed.

With *dach* [81] also being expressed in one of the *opsin2/opsin4* positive cell type families, we found all genes of the pax-six-eya-dach network to be expressed in opsin-expressing photoreceptors in the mature rudiment of *P. lividus*. 

### 4.3. Spatial and Temporal Analysis of Retinal Gene Products

#### 4.3.1. Pax6, Six3, Otx

In line with previous findings by Ullrich-Lüter et al. [16] in juvenile *S. purpuratus*, we found *pax6* highly expressed in the area where the developing podia emerge in mature rudiment of *P. lividus*. Later on, *pax6* and *six3* were expressed in secondary podia as they developed. In line with the findings of Byrne et al. [24] in *Heliocidaris erythrogramma*, a sequential up-regulation of those genes was observed when new podia formed, with a simultaneous down-regulation of expression in older podia. This cyclical pattern suggests a role for early specification of both genes in sea urchin PRC progenitors in podia.

Although *pax6* expression has repeatedly been reported to be closely associated within opsin expressing PRCs in adult sea urchin podia, it could so far not be co-localized on a cellular level. Agca et al. [14] reported *pax6* expression in adult *S. purpuratus* podial discs associated with parts of the nervous system and putative ciliated primary sensory cells. Lesser et al. [15] found *pax6* protein inside pigment cell nuclei of adult *Strongylocentrotus franciscanus* podial discs. Ullrich-Lüter et al. [16] showed that the clusters of ciliated cells at the periphery of *S. purpuratus* podial discs indeed host Opsin4-positive PRCs extending axons into the basiepithelial nervous system of the animal. However, as none of the aforementioned authors found co-expression of *pax6* and Opsin4 at the cellular level, this poses the question if differentiated photoreceptors in sea urchins retain *pax6* expression. Belecky-Adams et al. [82] report that *pax6* is initially broadly expressed in retinal progenitor cells of embryonic chickens, but later becomes restricted to prospective ganglion, amacrine and horizontal cells, while Hsieh & Yang [83] investigating the same tissue found a significant down-regulation of *pax6* protein as retinal progenitors undergo the preneurogenic to neurogenic transition. Our findings of *pax6* and *opsin4* being co-expressed in a small subset of cells, as indicated by scRNA-seq and of *pax6* downregulation in the developing sea urchin podia might thus reflect this transitionary process.

*Otx/orthodenticle* and members of the six family are also involved in the early specification of bilaterian PRCs [1]. Interestingly, scRNA-seq revealed *otx* as well as *six1/2* and *six3* expression inside one or several of the *opsin2/opsin4* positive cell type families, but not in all of them. This finding might represent different expression profiles of the same cell type caught at different stages of differentiation. Examining the spatial expression of *six3* inside the developing podia of the mature rudiment of *P. lividus*, we found a cyclical expression pattern resembling that of *pax6* and *six3* is early expressed on the oral side of the developing podia where the podial nerve and the podial ganglion reside, and its expression fades once the podia are fully differentiated.

#### 4.3.2. Opsins

Sea urchin podia contain a large amount of putative receptor cells [13] and have been demonstrated to express almost all opsin types found in sea urchins [9,10,14,15,16]. Opsin and retinal gene expression are clearly not restricted to podia (there is, e.g., opsin expression in spines and pedicellariae as well as in the integument) but most opsin types are expressed in podia, often with their highest expression levels in the podial disc, correlating with its high numbers of putative receptor cells. Due to the apparent lack of membrane enlargements or screening pigments [16] as typical for many PRCs in other bilateria [84], all those different opsins are presumably expressed in receptor cell types lacking membrane specializations. Our current findings on opsin and *pax6* expression inside the podia of *P. lividus* developmental stages support the view that more putative primary receptor cell types in the podial disc might have a photoreceptive function than previously thought.

The spatial expression patterns of Opsin4 protein and mRNA as well as of Opsin1 protein in developmental stages of *P. lividus* (Figure 7) corroborated expression patterns known from its sea urchin relative *S. purpuratus* [16]. The characterization of cells expressing Opsin2 is the first evidence of spatial expression of this gene in sea urchin mature rudiments and juveniles (Figure 7) and future research is needed to elucidate their potential function.

Clearly, we see a developmental dynamic regarding opsin expression between the mature rudiment stage and the juvenile. Whereas in the rudiment only two opsins are expressed (*opsin4* and *opsin2*), the juvenile shows a greater opsin diversity expressing all opsin types known in sea urchins except for sea urchin peropsin [18], the lack of which may well be due to the current state of the *P. lividus* genome annotation.

### 4.4. Sea Urchin Photoreceptors

Numbers of disc and basal PRCs are growing through development. While in the adult podia, the basal Opsin4 positive PRC clusters consist of more than 50 receptor cells each and average numbers of disc PRCs range between two and three digits [85], the primary podia of the rudiment and juvenile respectively showcase a very small number of opsin-expressing PRCs. However, as Opsin4-expressing PRCs are extremely light sensitive [86], a functional role of those PRCs e.g., in substrate evaluation prior to metamorphosis seems possible. So far, it seems that sea urchins do not perform directed phototaxis until they reach a certain age, which in *S. purpuratus* is approximately 2 months [16]. Based on their findings on the positioning of the adult sea urchins’ PRCs in skeletal depressions at the base of the podia, the authors have proposed those skeletal shapes function as light screening device and that the formation of the skeleton is thus a requirement for the animals’ ability to perform phototaxis. However, in terms of measuring the overall illumination level and especially how much light is actually reaching a specific spot inquired for metamorphosis, even a small number of Opsin4-positive PRCs lacking a screening element would be sufficient.

So far, we do not know if the Opsin4-expressing PRCs at the base of the podium and those within the disc share a similar morphological and/or molecular cell type identity. The immunohistological data on the two cell clusters suggest that both not only share some polarization in morphology, but also that their apical cell parts are enlarged, most presumably as microvillar extensions. As has been shown in *S. purpuratus*, the apical region of sea urchin tube foot disc Opsin4-positive PRCs indeed shows microvilli [16], although not as numerous and structurally organized as, e.g., in the rhabdome of insect eyes [87]. Further investigations, including ultrastructural studies, are essential to reveal the morphological identity of those cell types. However, our scRNA-seq data revealing two distinct sub-clusters of cell type families for *opsin2/opsin4* positive PRCs hints at two molecularly different clusters. This finding is especially interesting regarding the potentially different functions of those PRC clusters. While for the PRC cluster at the base of the tube foot, a functional role in spatial vision seems conceivable, the disc PRCs being part of highly motile appendages and lacking any light-screening pigment seem less likely to possess such abilities. The question of whether the two PRC cell type family clusters might have evolutionarily diverged and potentially been adapted to sense different photosensory stimuli (e.g., spatial vision vs. overall illumination level) needs to be investigated by further computational analysis of genomic/transcriptomic data, as well as by ultrastructural analysis of similarities and differences of the integration of those PRC clusters into the nervous system of the animals. Finally, molecular, behavioral, and electrophysiological data coming from animals reared under different light conditions and water parameters could contribute to deciphering the relationship between the opsins expressed and the environmental stimuli and conditions.

## Figures and Tables

**Figure 1 cells-11-02636-f001:**
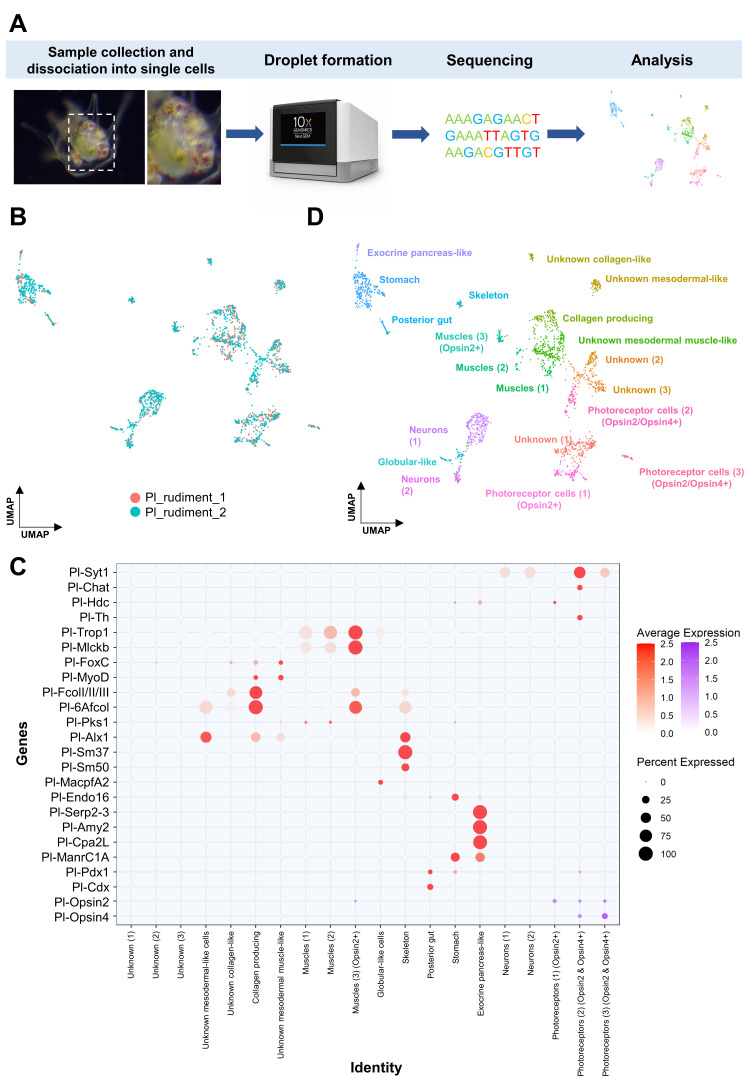
Cell type atlas of the *P. lividus* mature rudiment. (**A**) ScRNA-seq pipeline from larvae collection, rudiment dissection and dissociation to 10× capturing and computational analysis. *(***B**) Overlay UMAP showing the overlap of the libraries originating from the two biological replicates. (**C**) Dotplot showing the average expression of genes used as markers to identify specific cell clusters. (**D**) UMAP showing mature rudiment cell types, colored by their assignment to the initial set of 20 distinct cell clusters. Average expression gradient for opsin genes is depicted in purple, and for the rest of the genes, in red.

**Figure 2 cells-11-02636-f002:**
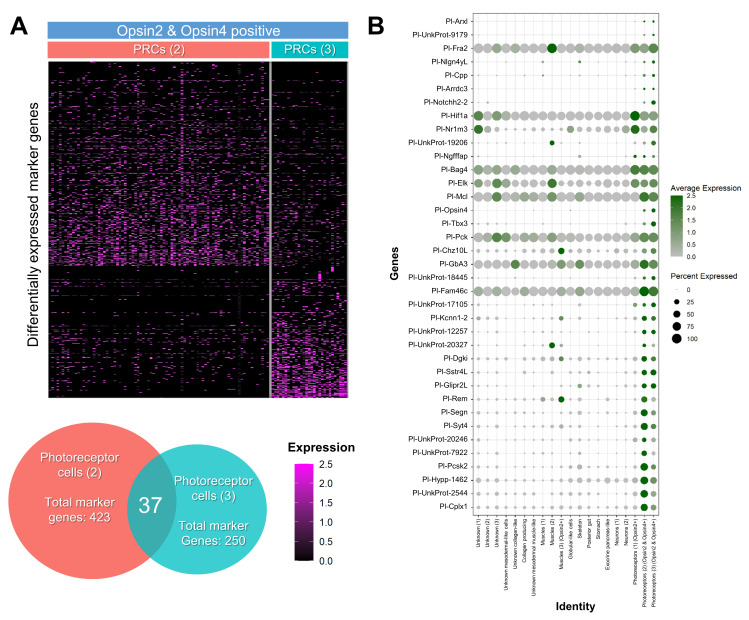
Photoreceptor cell type families and retinal molecular signature. (**A**) Heatmap and Venn diagram showing the differentially expressed marker genes between the two *opsin2*/*opsin4* double positive clusters (PCRs 2 and PCRs 3). (**B**) Dotplot showing the averaged scaled expression of common marker genes of the two *opsin2*/*opsin4* double positive clusters against the whole *P. lividus* mature rudiment atlas.

**Figure 3 cells-11-02636-f003:**
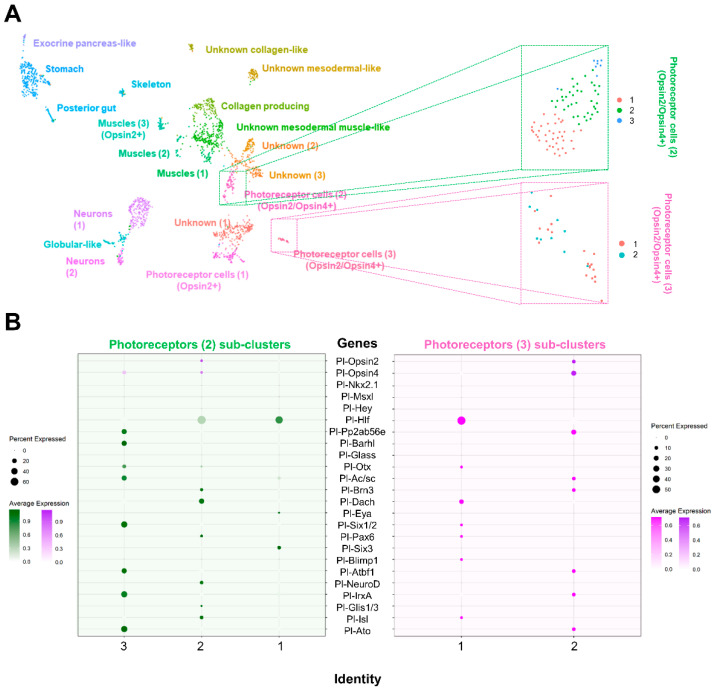
Distinct molecular signature of the two *opsin2*/*opsin4* double positive cell type families. (**A**) Sub-clustering analysis of the two *opsin2/opsin4* double positive cell type families resulted in the generation of the three and two distinct sub-clusters, respectively. (**B**) Dotplot showing the average scaled expression of sea urchin orthologues of retinal genes distributed in the different sub-clusters. Average expression gradient for opsin genes is depicted in purple, while the rest of the genes expressed in PRCs 2 and PRCs 3 sub-clusters are shown in green and magenta, respectively.

**Figure 4 cells-11-02636-f004:**
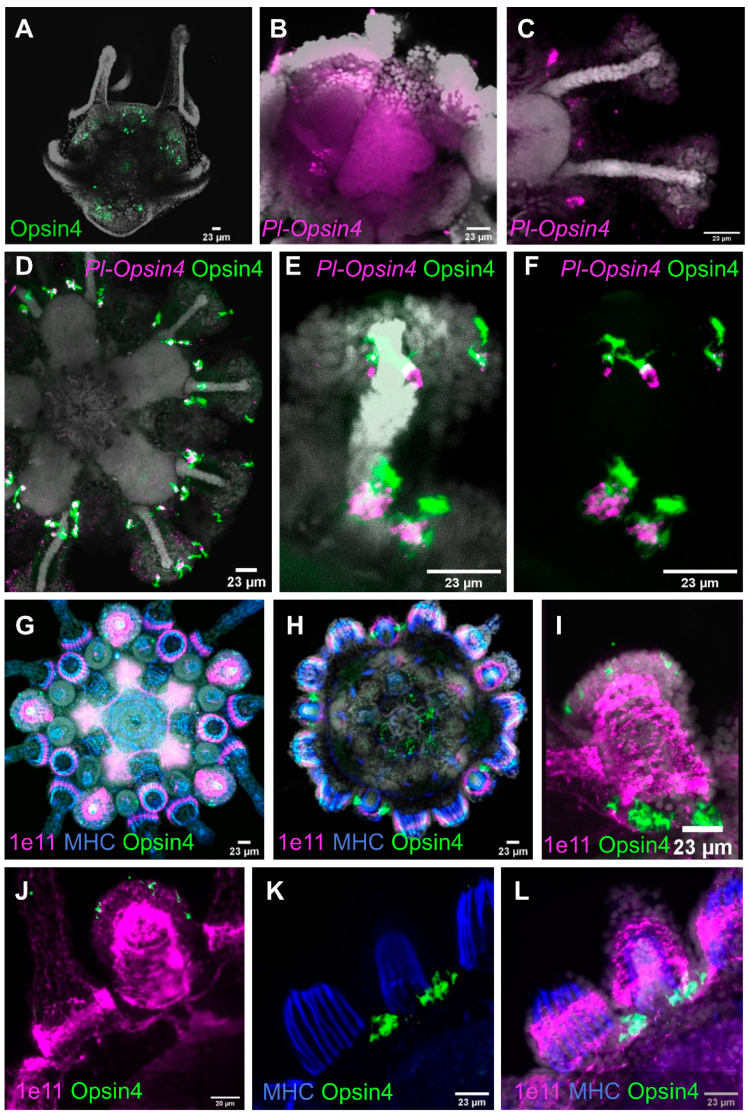
Immunolocalization (IHC) of sea urchin Opsin4 and expression of *opsin4* mRNA (FISH) in mature rudiments and juveniles. (**A**) Immunohistochemical detection of Opsin4 at the 8-arm mature rudiment stage. (**B**) FISH using an antisense probe against *Pl-opsin4* at the mature rudiment stage. (**C**) Immunohistochemical detection of Opsin4 protein at juvenile stage. (**D**–**F**) FISH using a specific antisense probe for *Pl-opsin4* paired with IHC for Opsin4 protein at juvenile stage. (**G**,**H**) IHC detection of antigens indicative of the nervous system (1e11), musculature (MHC) and PRCs (Opsin4). (**G**,**H**) are a result of compilation of different stacks corresponding to the same individual. (**I**) Podium close-up showing the immunolocalization of Opsin4 and the pan-neuronal marker 1e11. (**J**–**L**) IHC detection of antigens indicative of the nervous system (1e11), musculature (MHC) and PRCs (Opsin4) focusing on a podium and the surrounding spines. DAPI was used to visualize nuclei (gray). Orientation: (**A**), the specimen is viewed from the top; (**B**), the specimen is viewed from the side; (**D**,**G**,**H**), juveniles are in oral view.

**Figure 5 cells-11-02636-f005:**
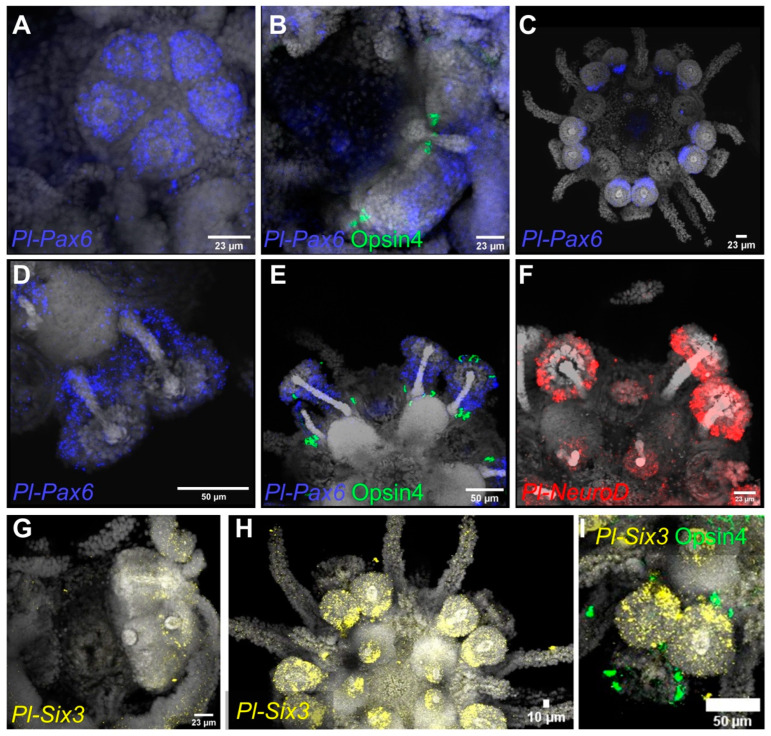
Retinal transcription factors *pax6*, *neuroD1* and *six3* RNA: expression in whole mount mature rudiments and juveniles. (**A**–**E**) FISH using a specific antisense probe against *Pl-pax6* at rudiment (**A,B**) and juvenile (**C**–**E**) stages. In B and E, FISH are also paired with IHC detection of Opsin4. (**F**) FISH detection of the *neuroD* transcripts at juvenile stage. (**G**–**I**) FISH using an antisense probe for *six3* at rudiment (**G**) and juvenile (**H**,**I**) stages. In I, FISH is paired with IHC for Opsin4. DAPI was used to visualize nuclei (gray). Orientation: (**A**), the specimen is viewed from the top; (**B**,**G**) the specimens are viewed from the side; (**C**–**F**,**H**,**I**) juveniles are in oral view.

**Figure 6 cells-11-02636-f006:**
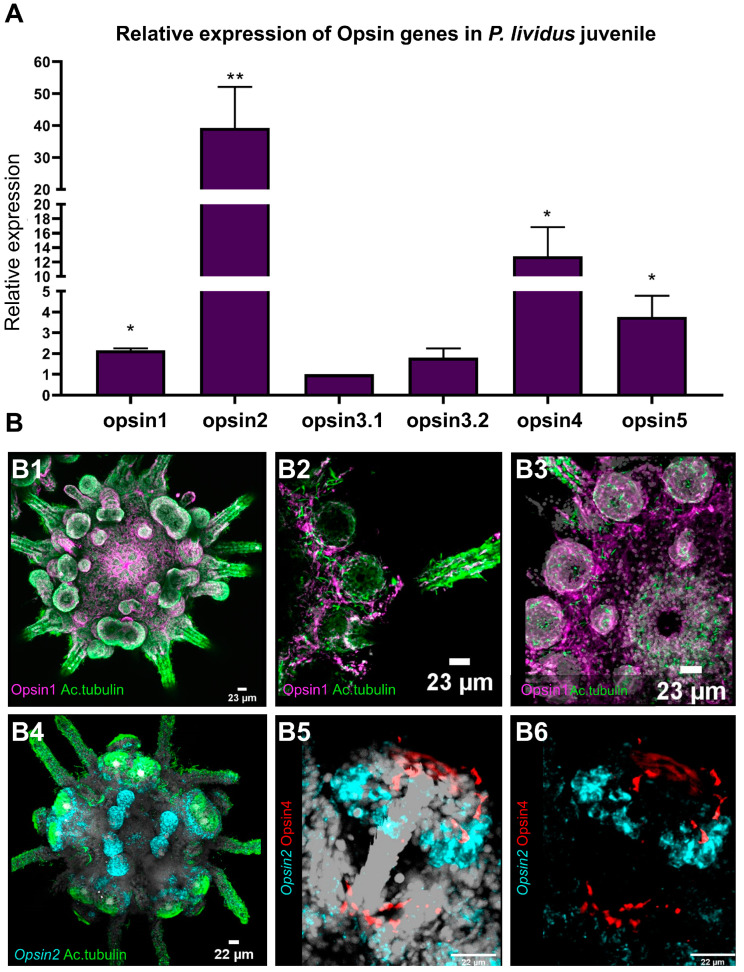
Expression analysis of Opsins in *P. lividus* juveniles. (**A**) Relative gene expression of different opsins in juveniles as revealed by qRT-PCR. Sample size: 3 biological replicates; Statistical significance cut-off criteria: * *p* < 0.05; ** *p* <0.01. (**B**) Gene expression visualization of Opsin1, *opsin2* and Opsin4 at juvenile stage. (**B1**–**B3**) double IHC detection of Opsin1 (magenta) and acetylated tubulin labeling ciliated structures (green). (**B4**) FISH using a specific antisense probe for *Pl-opsin2* (cyan) paired with IHC for acetylated tubulin (green). (**B5**,**B6**) FISH for Pl-Opsin2 (cyan) transcripts paired with IHC detection of Opsin4 (red). DAPI was used to visualize nuclei (gray).

**Figure 7 cells-11-02636-f007:**
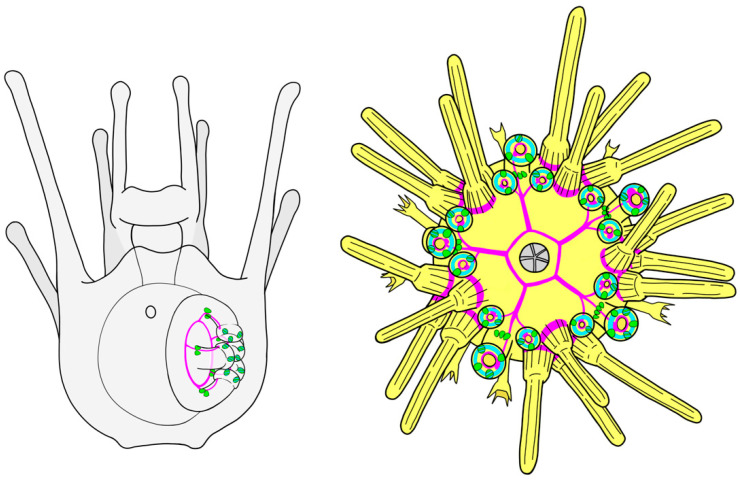
Schemes with the opsin expression on them and the molecular signature of the 2 clusters of PRC identified at mature rudiment stage. Nervous system in magenta, Opsin4+ PRCs in green, opsin2+ PRCs in cyan, Opsin1+PRCs in yellow.

## Data Availability

The data associated with the findings of this study can be found at Gene Expression Omnibus (NCBI GEO) [88] under the GEO Series accession number GSE211842 (https://www.ncbi.nlm.nih.gov/geo/query/acc.cgi?acc-=GSE211842, accessed on 25 July 2022).

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
