# Peer review of "A New Model Organism to Investigate Extraocular Photoreception: Opsin and Retinal Gene Expression in the Sea Urchin Paracentrotus lividus"

_cells, 2022, doi:10.3390/cells11172636_

Round 1

Reviewer 1 Report

- **A brief summary** - Sea urchins are animal models to learn about the evolution of extraocular photoreceptors. To understand the purple sea urchin's photosensitive system, based on the novel *Paracentrotus lividus* Genome (ref.31:under submission) and the single cell transcriptomes, Dr. Periklis Paganos and his colleagues investigated the extraocular photoreceptor cell system. The compositions and expression levels of the conserved opsin and retinal genes were determined to define the cell types of different developmental stages. In conclusion, a regulatory toolkit with a core of vertebrate retinal gene orthologs that controls photoreceptor specification and function of the sea urchins was given, and their tube feet as photosensory organs were confirmed.   - **General concept comments** - I definitely agree that the improvement of the P. lividus genome annotation should be essential to this study, which the authors have pointed out in line 582. More background about the opsin family, even the retinal orthologs, is needed for this research. Suggest the authors supply a supplement table. Personally, I wonder how about the rx and opsin3.2 reported in "Valencia, J.E.*et al.*(2021) ‘Ciliary photoreceptors in sea urchin larvae indicate pan-deuterostome cell type conservation’,*BMC Biology*, 19(1), p. 257. Available at:https://doi.org/10.1186/s12915-021-01194-y."   - **Specific comments**referring to line numbers, tables or figures that point out inaccuracies within the text or sentences that are unclear. These comments should also focus on the scientific content and not on spelling, formatting or English language problems, as these can be addressed at a later stage by our internal staff. - There are many unclear sentences that have affected my understanding of the scientific content. - Type 1: the long and frequent annotations in "( )" broke the sentences, such as in lines 39, 41, 56, 65, and so on. - Type 2: too many hyphens in a word we don't want to break across lines, such as in lines 2, 3, 13, and so on. You can turn off automatic hyphenation for the paragraph. You can also keep words or characters (numbers or letters) together in Microsoft Word using nonbreaking spaces or nonbreaking hyphens. - Type 3: spelling and formatting errors. - Lines 158-162, the Calcium- and magnesium-free artificial seawater (CMF-ASW) refers to the Cold Spring Harbor Protocols on website http://cshprotocols.cshlp.org/content/2009/12/pdb.rec12053. Of course, also accepted if use the spelling in the authors' previous article published in eLife https://elifesciences.org/articles/70416. Furthermore, 0,05% should be 0.05%; 0,22 should be 0.22. - Lines 178-184, Variance stabilizing transfer (VST) , Sharing Nearest Neighbor (SNN), minimum percent (min.pct) are suggested. - Line 202, "for ON" should be modified. - Line 403, double-localization still should be spelled as co-localization. - In Figures 1 & 3, the scales for the average expression levels of opsin genes should be depicted or deleted. I feel they are not necessary, at least in figure 1 and the right part of figure 3b. - In the legend of Figure 6A, the sample sizes (n=3) and the paired t-test with α=0.05 significant level are suggested to be added.   - Nonetheless, I think that this paper has the potential to attract the attention of biologists. It is worthy of consideration by Cells after a revision.

Author Response

Response to Reviewers

Before we address in detail the points raised by the two reviewers we would like to express our gratitude for their comments/suggestions that resulted in improving our manuscript. Based on the reviewers’ comments addressing issues found within the current manuscript, it is evident that they are experts in diverse fields. With our revisions, we worked to bridge the gap between reviewer#1 who appears to be an expert in opsin phylogeny and asks for more specific classification and concise referring to opsin genes exclusively by numbers and reviewer#2, who represents the more general reader in this specific context and, as of our understanding, would prefer the use of "functional" opsin gene names, as e.g. rhabdomeric vs. ciliary in order to enhance clarity of classification. To address both standpoints, we introduced opsin genes by their functional as well as numerical classification in the introduction, pointed out, in the results section, the phylogeny we used to classify the found opsin genes and from there on consistently stuck with opsin gene classification by numbers. More information on the investigated opsins and retinal genes can be found in the supplementary file 2, as suggested by reviewer#1.

Response to Reviewer#1

- **General concept comments** -

I definitely agree that the improvement of the P. lividus genome annotation should be essential to this study, which the authors have pointed out in line 582. More background about the opsin family, even the retinal orthologs, is needed for this research. Suggest the authors supply a supplement table. Personally, I wonder how about the rx and opsin3.2 reported in "Valencia, J.E.*et al.*(2021) ‘Ciliary photoreceptors in sea urchin larvae indicate pan-deuterostome cell type conservation’,*BMC Biology*, 19(1), p. 257. Available at:https://doi.org/10.1186/s12915-021-01194-y."  

We would like to start by thanking the reviewer for the very positive feedback, comments and recommendations that contributed to improving our manuscript. We completely agree that the improvement of the P. lividus genome annotation is essential to fully address the molecular signatures and gene regulatory wirings of this sea urchin species and in the future will be a great resource for the echinoderm and evodevo community. Regardless of the current status of the annotation, it is still outstanding the amount of information that can be extracted from omics data that are using it. However, we do share the same hope that soon the annotation will allow even deeper evodevo comparisons and characterization of cell types. We followed the reviewer’s recommendation and expanded the background of the opsin families, including a more detailed and organized description of the different opsin groups found in sea urchins. Regarding the retinal orthologs, we added several clarifications throughout the text. Moreover we thank the reviewer for suggesting the incorporation of a supplementary file (Supplementary file 2) in which more information on the retinal orthologs is summarized. Following the reviewer’s advice we generated a file in which the P. lividus gene ID, the gene name, the S. purpuratus gene ID, their expression profiles (according to the data shown in our study) and relevant publications on their role during vertebrate PRC development are presented.

Regarding the Rx gene, this is a very interesting observation and we are grateful to the reviewer for noticing that we’ve missed this reference. The Rx gene is not found to be expressed in our P. lividus transcriptomic data and this is very interesting since it has been previously shown in S. purpuratus larvae that the Rx gene is a member of an evolutionary deuterostomian conserved PRC gene regulatory wiring. We have incorporated the following part in the discussion to address the absence of rx in the PRCs found in mature rudiments as well as possible explanations on why it is not found/expressed.

“Previous studies carried out on S. purpuratus larvae demonstrated the presence of an evolutionary conserved gene regulatory module consisting of the transcription factors rx, otx, six3 and tbx2/3 in Go-opsin positive PRCs (61). Surprisingly, only otx transcripts were found in opsin4 positive clusters, while neither rx nor tbx2/3 transcripts were detected in any of the PRCs analyzed in this study. These data suggest that the sea urchin Go-opsin positive photoreceptors and rhabdomeric photoreceptors follow diversified genetic programs. We cannot, however, rule out the potential that these transcription factors are active during PRC differentiation or specification at developmental times other than the ones sampled in this work. Therefore, we believe future studies are needed to reach a safe conclusion regarding the spatiotemporal expression of the genes involved in the rhabdomeric PRCs gene regulatory network.”

- **Specific comments**referring to line numbers, tables or figures that point out inaccuracies within the text or sentences that are unclear. These comments should also focus on the scientific content and not on spelling, formatting or English language problems, as these can be addressed at a later stage by our internal staff. -

  1. There are many unclear sentences that have affected my understanding of the scientific content. - Type 1: the long and frequent annotations in "( )" broke the sentences, such as in lines 39, 41, 56, 65, and so on. - Type 2: too many hyphens in a word we don't want to break across lines, such as in lines 2, 3, 13, and so on. You can turn off automatic hyphenation for the paragraph. You can also keep words or characters (numbers or letters) together in Microsoft Word using nonbreaking spaces or nonbreaking hyphens. - Type 3: spelling and formatting errors.

We apologize to the reviewer for this. We have modified the manuscript reducing the frequent annotations in parentheses, eliminated hyphens and fixed spelling and formatting errors.

  1. Lines 158-162, the Calcium- and magnesium-free artificial seawater (CMF-ASW) refers to the Cold Spring Harbor Protocols on website http://cshprotocols.cshlp.org/content/2009/12/pdb.rec12053. Of course, also accepted if use the spelling in the authors' previous article published in eLife https://elifesciences.org/articles/70416. Furthermore, 0,05% should be 0.05%; 0,22 should be 0.22. - Lines 178-184, Variance stabilizing transfer (VST) , Sharing Nearest Neighbor (SNN), minimum percent (min.pct) are suggested.

We thank the reviewer for spotting this. We have adapted the abbreviation according to our previously published eLife article. Decimals have been amended and Variance stabilizing transfer (VST) , Sharing Nearest Neighbor (SNN), minimum percent (min.pct) were added.

  1. Line 202, "for ON" should be modified.

We apologize for this. “For” has been eliminated and replaced with incubated overnight (ON) at 4°C

  1. Line 403, double-localization still should be spelled as co-localization.

Double localization has been replaced with co-localization.

  1. In Figures 1 & 3, the scales for the average expression levels of opsin genes should be depicted or deleted. I feel they are not necessary, at least in figure 1 and the right part of figure 3b.

We appreciate the reviewers comment, however we believe that their presence is necessary to more general readers who might be less familiarwith this technique, in order to better understand the figures. We would appreciate it if we could keep it as it is.

  1. In the legend of Figure 6A, the sample sizes (n=3) and the paired t-test with α=0.05 significant level are suggested to be added.   

We agree with the reviewer on this. Sample size and cut off significance values were added in the figure legend.

Reviewer 2 Report

Introduction

L127 – L140: Perhaps, in the last paragraph, the authors could make clearer the objectives of the study.

Material and methods

L143 – L151: Since the study focus on light sensitive proteins, it would be interesting to add the light conditions/regime while rearing these larvae/juveniles as well as the dimensions of the aquaria. This might be an important information for future studies comparing single cell seq RNA data and expression levels of these opsins at the same developmental stage in this species, different stages, or close species. Is there any information regarding the water parameters besides the temperature?

L167 – L188: Do you have the assembled sequences of the so-called “specific echinopsins”? Comparing them to other animal opsins might be interesting.

L211 – L224: Were these profiles obtained using the animals reared under the same conditions as of the topic 2.1?

Results

L227 – L235: This paragraph evokes the thought that the study is focused on the opn4 but according to the last paragraph of the Introduction it seems that the objective is broader than covering the opn4 only. Also, as a suggestion, this paragraph could be reduced since most of it is referring to the methods.

Fig. 1: The title of the figure is referring to P. lividus pre-metamorphosed juvenile but in 2.2 and Fig. 1A description P. lividus samples are referred as larvae. As I understood, the mature rudiment stage was used for the single cell transcriptomics and the juveniles for the qPCR. If so, please be consistent when referring to the stage used in each method; that helps the reader to follow your reasoning and avoid misunderstanding. What the colors of the average expression gradient bars represent? Different biological replicates or the opsin presence? Please, specify.

L271 – L274: The “Muscles (3) (Opsin2 +)” type is not included here?

Fig. 2: I am not sure if it is related to the reviewer version, but the resolution of this figure seems to be lower when compared to the Fig. 1.

L408 – L417: The classification of the animal opsins is sometimes very confusing. When the authors are referring to different opsins using “opsin1”, “opsin2” etc., are these the same as of previous classifications? According to the introduction (L70 – 71), for example, apparently the authors are referring to the general classification of the animal opsins. That could confuse the reader when it is stated (L413) “opsin2, an echinoderm-specific opsin”, or (L415), “Another echinoderm-specific opsin, opsin5”. Different studies do not necessarily follow the exact same classification, which could make difficult future comparisons. It might be helpful a phylogenetic tree containing some reference opsins and the opsins found in this study that could guide the reader through this reasoning.

Discussion

L443 – L445: The Fig. 1 title refers to “pre-metamorphosed juvenile” while Fig 6 refers to “P. lividus juveniles”. So, the only techniques comparing different stages concomitantly are IHC and FISH?

L574 – L576: The name of this opsins is just “opsin2”? That could confuse the reader because of previous classifications of the animal opsin group.

L580 – L583: Here there is a comparison between scRNA-seq and qRT-PCR data in two different developmental stages. The differences are interesting, but it would be even more interesting if this comparison would have been made between different stages using the same methods. It is also important to know if the specimens were reared under the same light conditions as well as if them were obtained under the same conditions to avoid bias. I know that is not necessarily the objective of the study, but that would be a very useful information.

L614 – L619: A future comparison rearing these specimens under different light conditions/water parameters would also be helpful to demonstrate the relation between these opsins and environmental conditions. As mentioned in the text, we still lack information about the functions of many of these opsins.

Author Response

Response to Reviewers

Before we address in detail the points raised by the two reviewers we would like to express our gratitude for their comments/suggestions that resulted in improving our manuscript. Based on the reviewers’ comments addressing issues found within the current manuscript, it is evident that they are experts in diverse fields. With our revisions, we worked to bridge the gap between reviewer#1 who appears to be an expert in opsin phylogeny and asks for more specific classification and concise referring to opsin genes exclusively by numbers and reviewer#2, who represents the more general reader in this specific context and, as of our understanding, would prefer the use of "functional" opsin gene names, as e.g. rhabdomeric vs. ciliary in order to enhance clarity of classification. To address both standpoints, we introduced opsin genes by their functional as well as numerical classification in the introduction, pointed out, in the results section, the phylogeny we used to classify the found opsin genes and from there on consistently stuck with opsin gene classification by numbers. More information on the investigated opsins and retinal genes can be found in the supplementary file 2, as suggested by reviewer#1.

Response to Reviewer#2

We would like to start by thanking the reviewer for the insightful feedback and the constructive comments/suggestions that resulted in a substantial improvement of the manuscript. Overall we’ve followed most of the reviewer’s suggestions and we’ve adjusted the manuscript accordingly.

  1. L127 – L140: Perhaps, in the last paragraph, the authors could make clearer the objectives of the study.

We thank the reviewer for this suggestion. We have modified the last paragraph so that the objectives and our findings are clearer.

  1. L143 – L151: Since the study focus on light sensitive proteins, it would be interesting to add the light conditions/regime while rearing these larvae/juveniles as well as the dimensions of the aquaria. This might be an important information for future studies comparing single cell seq RNA data and expression levels of these opsins at the same developmental stage in this species, different stages, or close species. Is there any information regarding the water parameters besides the temperature?

We apologize to the reviewer for not including this information, which we agree is very important for such studies as ours. We’ve included further details on the light conditions, the salinity of the Mediterranean Sea sea water and the culturing containers. 

  1. L167 – L188: Do you have the assembled sequences of the so-called “specific echinopsins”? Comparing them to other animal opsins might be interesting.

We thank the reviewer for this comment and we think there might be a misunderstanding. D’Aniello and colleagues (ref:19) have performed phylogenetic comparisons addressing Opsin evolution in Ambulacraria and the term echinopsins originates from the data of that study. We have adapted the manuscript accordingly so that the opsin classification is clearer as well as the classification we chose to use throughout the revised manuscript. 

  1. L211 – L224: Were these profiles obtained using the animals reared under the same conditions as of the topic 2.1?

We are grateful to the reviewer for this observation. Indeed the growing conditions were exactly the same as the ones described in section 2.1. We’ve modified the manuscript accordingly in order to make it more clear.

  1. L227 – L235: This paragraph evokes the thought that the study is focused on the opn4 but according to the last paragraph of the Introduction it seems that the objective is broader than covering the opn4 only. Also, as a suggestion, this paragraph could be reduced since most of it is referring to the methods.

We thank the reviewer for spotting this inconsistency. The paragraph has been modified accordingly and its length has been reduced by eliminating too technical explanations already included inside materials and methods.

  1. Fig. 1: The title of the figure is referring to P. lividus pre-metamorphosed juvenile but in 2.2 and Fig. 1A description P. lividus samples are referred as larvae. As I understood, the mature rudiment stage was used for the single cell transcriptomics and the juveniles for the qPCR. If so, please be consistent when referring to the stage used in each method; that helps the reader to follow your reasoning and avoid misunderstanding. What the colors of the average expression gradient bars represent? Different biological replicates or the opsin presence? Please, specify.

We apologize for the inconsistency in terminology, which we have fixed in the revised manuscript. Colors of the average expression gradient bars indicate expression levels of the genes plotted. We have added a detailed description of what the colors represent in figures 1 and 3.

  1. L271 – L274: The “Muscles (3) (Opsin2 +)” type is not included here?

We are sorry for not including this cell type family here. The text has been modified as follows:

Then, we set out to accomplish our main goal, which was to investigate the photoreceptor composition of the sea urchin P. lividus. To this end we took advantage of the opsin phylogeny performed by D’Aniello and colleagues [19] and classified all opsins found accordingly (Supplementary file 2). When all the opsin genes encoded in the sea urchin genome were plotted, only two opsin-type PRCs and in total four distinct cell type families were found in the mature rudiment stage (Fig. 1C). One of these families correspond to a cell population expressing Opsin2 and histidine decarboxylase (Photoreceptor cells 1), two of these families (Photoreceptor cells 2 and 3) correspond to neurons (syt1 positive) co-expressing opsin2 and opsin4, while the last one (Muscles 3) resembles a muscle-like cell cluster expressing exclusively opsin2.

  1. Fig. 2: I am not sure if it is related to the reviewer version, but the resolution of this figure seems to be lower when compared to the Fig. 1.

We thank the reviewer for this. Figure 2 has been updated and the resolution issue is fixed.

  1. L408 – L417: The classification of the animal opsins is sometimes very confusing. When the authors are referring to different opsins using “opsin1”, “opsin2” etc., are these the same as of previous classifications? According to the introduction (L70 – 71), for example, apparently the authors are referring to the general classification of the animal opsins. That could confuse the reader when it is stated (L413) “opsin2, an echinoderm-specific opsin”, or (L415), “Another echinoderm-specific opsin, opsin5”. Different studies do not necessarily follow the exact same classification, which could make difficult future comparisons. It might be helpful a phylogenetic tree containing some reference opsins and the opsins found in this study that could guide the reader through this reasoning.

We appreciate the reviewer’s comment with whom we agree. We modified the text accordingly, by providing a first description of opsin genes by their functional as well as numerical classification in the introduction, followed by the phylogeny information which we used to classify them in the results section. From that point and onwards we consistently used the opsin gene classification by (sea urchin) numbers.

  1. L443 – L445: The Fig. 1 title refers to “pre-metamorphosed juvenile” while Fig 6 refers to “P. lividus juveniles”. So, the only techniques comparing different stages concomitantly are IHC and FISH?

Once again we thank the reviewer for spotting inconsistencies in terminology. We have adapted the name of Fig1 to “mature rudiment”. Yes, the only techniques presented in this manuscript that are common between the two developmental stages are IHC and FISH. The main focus of the manuscript is on the photoreceptors present prior to metamorphosis and then the investigation of their distribution in the juveniles. We have adapted the text to make it clearer to the reader which techniques have been used on which stage as well as why we have decided to do so. We believe that while the stage specific data are solid, independently of the methods used, future studies are needed to compare the mature rudiment and juvenile stages.

  1. L574 – L576: The name of this opsins is just “opsin2”? That could confuse the reader because of previous classifications of the animal opsin group.

As mentioned above the naming of the opsins has been adapted to avoid reader’s confusion.

  1. L580 – L583: Here there is a comparison between scRNA-seq and qRT-PCR data in two different developmental stages. The differences are interesting, but it would be even more interesting if this comparison would have been made between different stages using the same methods. It is also important to know if the specimens were reared under the same light conditions as well as if them were obtained under the same conditions to avoid bias. I know that is not necessarily the objective of the study, but that would be a very useful information.

We do agree with the reviewer that it would be interesting to use the same methods especially sc-RNA seq to explore PRC development. Unfortunately, at the moment scRNA-seq on juveniles is challenging due to lack of juveniles and difficulties in dissociating them, mostly due to their calcified skeleton. We also agree that mentioning the light conditions used to rear both larvae and juveniles is needed to avoid bias and therefore we added this information. 

  1. L614 – L619: A future comparison rearing these specimens under different light conditions/water parameters would also be helpful to demonstrate the relation between these opsins and environmental conditions. As mentioned in the text, we still lack information about the functions of many of these opsins.

We could not agree more with the reviewer on this. We followed the reviewers’ train of thought and included the following part in the last paragraph of the revised manuscript.

“Finally, molecular, behavioral, and electrophysiological data coming from animals reared under different light conditions and water parameters could contribute to deciphering the relationship between the opsins expressed and the environmental stimuli and conditions.”